



**Recent decrease in summer precipitation over the Iberian Peninsula**
**closely links to reduction of local moisture recycling**
Yubo Liu[1,2], Monica Garcia[3], Chi Zhang[1,4], Qiuhong Tang[1,2*]
[1]Key Laboratory of Water Cycle and Related Land SurfaceProcesses, Institute of Geographic
Sciences and Natural Resources Research, Chinese Academy of Sciences, Beijing, China
[2]University of Chinese Academy of Sciences, Beijing, China
[3]Department of Environmental Engineering, Technical University of Denmark, Lyngby, 2800,
Denmark
[4]Key Laboratory of Land Surface Pattern and Simulation, Institute of Geographic Sciences and
Natural Resources Research, Chinese Academy of Sciences, Beijing, China
**Correspondence**: Qiuhong Tang (tangqh@igsnrr.ac.cn)



## Abstract

The inherently dry summer climate of the Iberian Peninsula (IP) is undergoing drought exacerbated by more intense warming and reduced precipitation. Although many studies have studied changes in summer climate factors, it is still unclear how the changes in moisture contribution from the source lead to the decrease in summer precipitation. This study investigates the differences in the IP precipitationshed between 1980-1997 and 1998-2019 using the Water Accounting Model-2layers with ERA5 data, and assesses the role of local recycling and external moisture in reducing summer precipitation. Our findings indicate that the moisture contributions from the local IP, and from the west and the east of the precipitationshed contributed 1.7, 3.6 and 1.1 mm mon$^{-1}$ less precipitation after 1997 than before 1997, accounting for 26 %, 57 % and 17 % of the main source supply reduction, respectively. The significant downward trend of the IP local recycling closely links to the disappearance of the wet years after 1997 as well as the decrease of local contribution in the dry years. Moreover, the feedback between the weakened local moisture recycling and the drier land surface can exacerbate the local moisture scarcity and summer drought.



## 1. Introduction


The Iberian Peninsula (IP) is located in the Mediterranean area, which is among
the global "hotspots" of climate change. The IP precipitation is characterized by the
diverse climatic regimes and high spatial variability as a consequence of its geographic
position between the Atlantic Ocean and the Mediterranean Sea and its orographic
configuration. In responding to climate change with frequent heatwaves and above-
average warming, the IP is experiencing widespread decreases in precipitation,
especially in summer (Brogli et al., 2019; Cramer et al., 2018; Rajczak and Schär, 2017).
This reduction in summer precipitation is a major driver of water resource depletion
and the evolution of drought (Lopez-Bustins and Lemus-Canovas, 2020; Páscoa et al.,
2021; Teuling et al., 2013). To clarify the reason for the decrease in summer
precipitation, it is necessary to explain the changes in moisture contribution from the
source, such as local recycling and external sources.
Analysis of source supply and transportation in the hydrological cycle has become
one efficient way to understand well regional precipitation. With the introduction of the
concept of precipitationshed (Keys et al., 2014; Keys P. W. et al., 2011), which better
reveals the contribution from upwind evaporation sources to precipitation in downwind
sink region, it is more scientific and systematic to explain the precipitation variations
by using the fluctuations of moisture contribution as a precursor. Given the importance
of studying the source of precipitation, that is, precipitationshed, a variety of methods
have been developed and adopted, including physical isotope analysis (Bonne et al.,





2014), and numerical analytical models, either online methods running in parallel with
climate models (Damián and Gonzalo, 2018; Stohl and James, 2004, 2005), or offline
"posteriori models" (van der Ent and Savenije, 2011; van der Ent et al., 2010; van der
Ent et al., 2013). Although the mechanisms of these studies are different, they all
emphasize that the constantly changing source-sink relationship of atmospheric
moisture is an essential part of climate change research as global change continues.

Gimeno et al. (2010) comprehensively investigated the atmospheric moisture

sources of the IP precipitation at different scales, and identified the tropical–subtropical
North Atlantic corridor, the surrounding Mediterranean Sea and the local IP as the
important moisture regions. The high precipitation in the cold season is mainly
dominated by westerly wind regimes. The mid-latitude atmospheric dynamics, such as
the baroclinic synoptic-scale perturbations from the Atlantic and the polar jet stream,
as well as the high moisture supply from an Atlantic "tropospheric river" seem to be
responsible for the abundant precipitation during the cold season (Cortesi et al., 2013;
Ulbrich et al., 2015; Zhu and Newell, 1998). Compared to the rainy winter, the summer
with very low precipitation receives less attention. The subtropical location under the
descending air extending from the North Atlantic subtropical high controls low summer
precipitation over the IP, and local convective events increase the importance of local
recycling during summer (Serrano et al., 1999). Accordingly, the summer IP
precipitation, a significant proportion of which is taken up by the local recycled water
vapor, is completely different from the precipitation in winter that is dominated by the



moisture transported over long distances from external sources.

In recent decades, the increasing severity of summer drought in the IP, which is

closely related to precipitation variations, has attracted more attention. Several
mechanisms, including soil-atmosphere interactions (Boé and Terray, 2014), cloud
processes (Lenderink et al., 2007; Tang et al., 2012) and large-scale circulation changes
(Boé et al., 2009; Brogli et al., 2019; Kröner et al., 2017), have been found to be
potentially important for this complex summer climate change, which also appear to
have an impact on precipitation reduction. However, there is still a lack of
understanding of such summer precipitation decline in terms of changes in the moisture
contribution from the source. Therefore, tracing the precipitationshed of the IP and
quantifying the moisture contributions can provide us with a new perspective to analyze
the changes in IP precipitation. This study aims to evaluate the moisture contribution
of local recycling and external sources to the reduction of IP summer precipitation.

## 2. Study Area, Data and Methods

### 2.1 Study area

The IP is located in southwestern Europe at midlatitudes of the northern

hemisphere. It covers Portugal and the mainland of Spain. The geographic location of
IP is shown in Fig. 1(a) (36°N-44°N, 10°W-3°E) in a transition zone between
midlatitude and subtropical atmospheric circulation regimes. It has a complex
topography, surrounded by the Atlantic Ocean and Mediterranean Sea, and high in the



middle and northeast. The topographic and coastal processes affect water vapor
transport, forming a spatial precipitation gradient from northwest to southeast.
Extracted from the land-sea mask provided by European Centre for Medium-range
Weather Forecasts (ECMWF), the red outline area composed of multiple single 1×1
degree grids is our study area of IP.

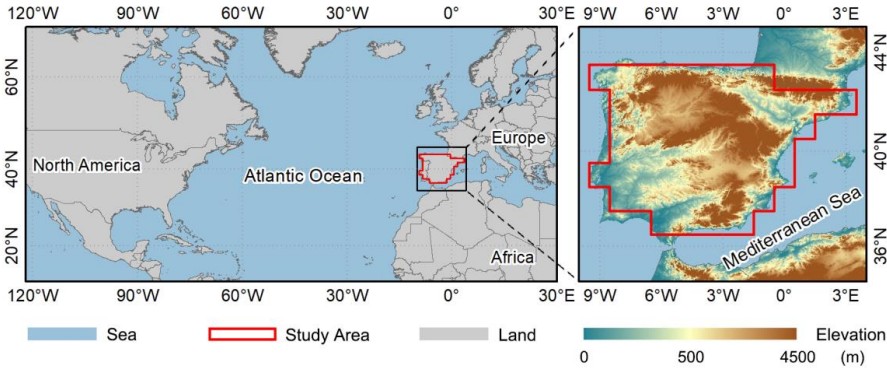

**Figure 1** Map of the IP (the area within the closure of the red line) on a grid of 1°×1° as the target
region.

2.2 Data

The newest reanalysis data held in ECMWF data archive, ERA5 dataset

downloaded from the Copernicus Climate Change Service (C3S) Climate Data Store
(CDS) is used in this study (Hersbach et al., 2020). The variables include surface
pressure, precipitation, evaporation, total column water, and vertical integrated
eastward and northward atmospheric water fluxes (including cloud liquid water flux,
cloud frozen water flux and water vapor flux) on single level, as well as the horizontal





U/V components of wind fields and specific humidity at the lowest $23^{rd}$ pressure levels
(200-1000hPa). The time resolution and spatial resolution selected for these data are 1
hour and 1×1 degree, respectively. This dataset covers the period from 1980 to 2019.
Compared to the old version reanalysis data (e.g., ERA-Interim or ERA-40), ERA5
combines vast amounts of historical observations into global estimates using more
advanced modelling and data assimilation systems (Hersbach et al., 2020).
To avoid the uncertainty of ERA5 precipitation as a global forecast data, its
reliability in the IP needs to be verified. Therefore, an observational gridded dataset
generated from a dense network of stations over the IP, named Iberia01 (Herrera et al.,
2019), is used to verify the accuracy of ERA5 precipitation data. Iberia01 provides the
daily precipitation for the period of 1971-2015 at 0.1×0.1 degree.
2.3 Model and methods
2.3.1 Water Accounting Model-2layers
Water Accounting Model-2layers (WAM-2layers) is an offline Eulerian method
tracking the moisture cycle forwards or backwards that quantifies the source-sink
relations (van der Ent et al., 2013; van der Ent et al., 2014). Its backward algorithm was
used in this study to trace the precipitationshed of the IP. The model of WAM-2layers
is an updated version of the original WAM. The water vapor balance equation in the
WAM-2layers algorithm maintains the premise that the atmosphere is well mixed, but
compared with the previous model, it takes the stratification of the atmosphere into



consideration. Thus, when the algorithm is applied to a specific region, the calculation
is as follows,
$$\frac{\partial W_{l,r}}{\partial t} + \frac{\partial (W_{l,r}u)}{\partial x} + \frac{\partial (W_{l,r}v)}{\partial y} = E_{l,r} - P_{l,r} \pm F_{V,r} + \alpha_{l,r} \quad (1)$$

where $W$ is the atmospheric moisture storage, or namely, precipitable water; $t$ is time; $u$
and $v$ are the wind components in $x$ (zonal) and $y$ (meridional) direction, respectively;
$E$ is evaporation; $P$ is precipitation; $F_V$ is the vertical moisture transport between the
bottom and top layer; $\alpha$ is a residual term; the subscript $l$ represents the portion in layer
$l$ (either the bottom layer or the top layer), and the subscript $r$ represents the tagged
portion provided by the source region.

Based on the assumption of a well-mixed atmosphere (Burde, 2010; Goessling and

Reick, 2013), the moisture contribution, that is, the tagged evaporation $E_r$, can be
calculated considering that the ratio of tagged to total atmospheric water storage is equal
to the ratio of tagged to total evaporation, as shown in Eq. (2). Considering the proposed
retention time of atmospheric moisture is about 1 week to 10 days (Numaguti, 1999),
we set the backtracking time as 1 month for summer precipitation to make sure that
more than 90 % of the precipitation can be redistributed to the surface.
$$E_r(t, x, y) = \frac{W_r(t,x,y)}{W(t,x,y)} \times E(t, x, y) \quad (2)$$

The main moisture source suppling IP summer precipitation, that is, 90[th] percentile

precipitationshed in this study, is divided into subregions to evaluate the role of the
contribution from each area, such as local recycling and external advection moisture.





For each of the partitioned source region ($A$), the proportion of the moisture
contribution from all grids covered by it to the total contribution from the main source
region ($MS$) is the contribution ratio ($CR$), which is calculated as the following Eq. (3).
The precipitation recycling ratio of the IP can be substituted with IP local contribution
ratio $CR_{IP}$.
$$CR_A = \frac{\sum E_r(t,x,y|A)}{\sum E_r(t,x,y|MS)} \times 100\% \quad (3)$$

### 2.3.2 Significance test

The slope significance of trend fitting and the significance of the difference in the
means are tested using Student t-test in this study. Additionally, the mutation analysis
for detecting significant mutation in precipitation series is the sliding t-test,
$$T = \frac{\frac{1}{n_1}\sum_{t=1}^{n_1} x - \frac{1}{n_2}\sum_{t=n_1+1}^{n_1+n_2} x}{\frac{(n_1-1)S_1^2+(n_1-1)S_2^2}{n_1+n_2-2}\sqrt{\frac{1}{n_1}+\frac{1}{n_2}}} \quad (4)$$

where $x$ is the precipitation series to be tested, $n_1$ and $n_2$ are step lengths set for two
sequences before and after the moving point, and $S_1^2$ and $S_2^2$ are the variances of the two
sequences which can be calculated as following.
$$S_1^2 = \frac{1}{n_1-1}\sum_{t=1}^{n_1}\left(x - \frac{1}{n_1}\sum_{t=1}^{n_1} x\right)^2 \quad (5)$$

$$S_2^2 = \frac{1}{n_2-1}\sum_{t=n_1+1}^{n_1+n_2}\left(x - \frac{1}{n_2}\sum_{t=n_1+1}^{n_1+n_2} x\right)^2 \quad (6)$$



## 3. Results

### 3.1 Evaluation and variation of precipitation data

The precipitation time series of ERA5 and Iberia01 data are shown in Fig. 2. The fluctuations and variations of ERA5 precipitation data are in good agreement with the observed data on both annual and seasonal scales, together with all correlation coefficients higher than 0.95. The average annual precipitation over the IP is about 55.66 mm mon$^{-1}$ from ERA5 and 58.07 mm mon$^{-1}$ from Iberia01, respectively. Compared with the observed data, the reanalysis data slightly underestimates the IP precipitation with the root mean square error (RMSE) of 3.46 mm mon$^{-1}$ on the annual scale. The comparison of seasonal precipitation shows that ERA5 is lower than the observed Iberia01 value in the rainy seasons (both winter and autumn), but higher in the dry summer. The RMSE between the two datasets of seasonal precipitation is in the range of 4.30-12.65 mm mon$^{-1}$. Since Iberia01 data is the grid data interpolated from observation site data (Herrera et al., 2019), some of the deviations between the ERA5 and Iberia01 precipitation can be partially affected by the interpolation process rather than solely the result of the error generated by the reanalysis process. In general, ERA5 precipitation data shows the characteristics of IP precipitation reasonably well and thus is suitable for studying the changes.



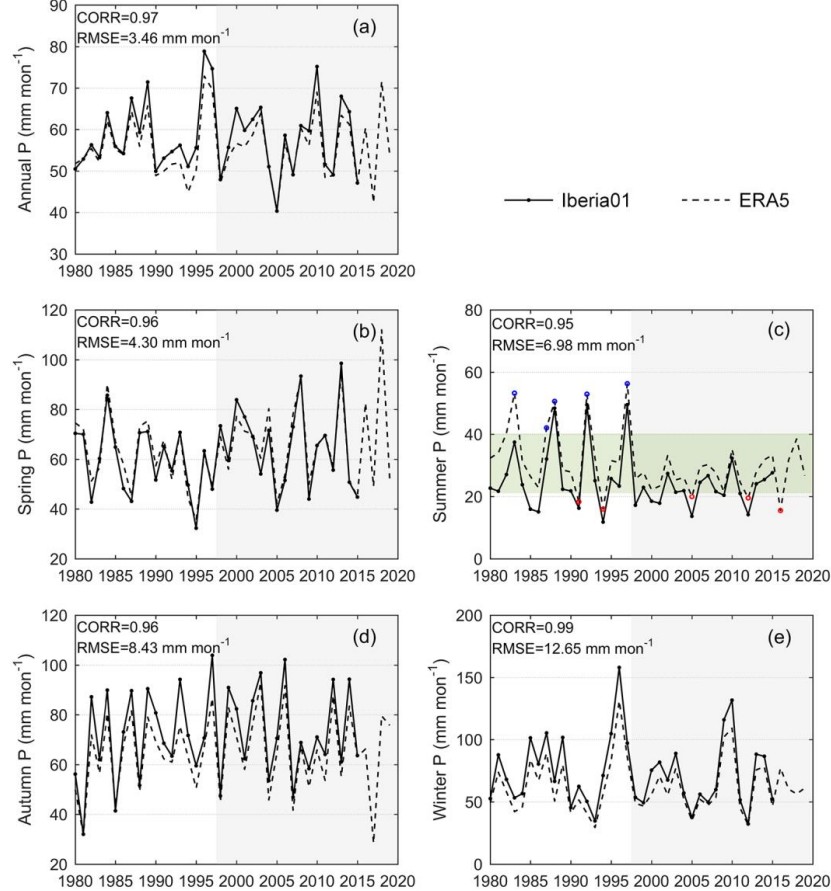

**Figure 2** Variations of IP annual precipitation (a), spring (March, April and May, b), summer (June,

July and August, c), autumn (September, October and November, d) and winter (December, January

and February, e) during 1980-2019. The green shading covers the interval of one standard deviation

of summer precipitation. The years with summer ERA5 precipitation exceeding the range of the

green shading interval are circled in blue and red.

Only in summer, the mutation analysis of the two sets of precipitation data,

Iberia01 and ERA5, both show statistically significant changes in 1997. Accordingly,



the entire 40-year period is divided into two periods, 1980-1997 and 1998-2019, to
compare the difference in summer precipitation between the two periods. The average
summer precipitation is 34.89 and 27.17 mm mon$^{-1}$ before and after 1997, respectively.
Compared with 1980-1997, the average summer precipitation during 1998-2019
decreases by 7.72 mm (22.13 %) in the whole study area. On the grid scale, almost all
grids have less precipitation after 1997, and more than half of all grids show the
statistically significant reductions (Fig. 3). However, this change is unevenly distributed
in space, as shown by the greater reduction in the grids on the northeastern IP that can
even exceed 10 mm mon$^{-1}$.

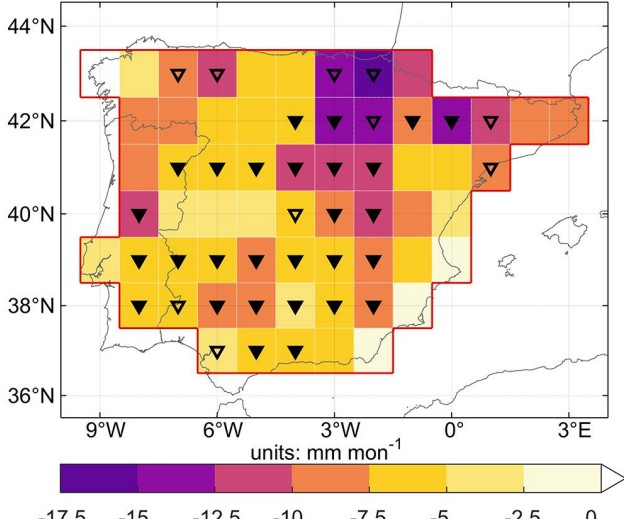


**Figure 3** The difference of average summer precipitation over the IP between 1998-2019 and 1980-
1997 (average of 1998-2019 minus average of 1980-1997). The triangles indicate the differences
are significant at 0.05 (solid) and 0.1 (hollow) level.





For summer precipitation, the dry years (1991, 1994, 2005, 2012 and 2016) and
the wet years (1983 1987 1988 1992 and 1997) are selected, which are circled in Fig.
2(c). A wet year is defined as the year in which the precipitation is more than one
standard deviation above the average precipitation, and similarly, the precipitation in a
dry year is lower than a standard deviation range. Accordingly, the division of time
period also applies to the precipitation series of the dry and wet years. It is specifically
observed that the dry years are separated, with the average precipitation of 17.15 and
18.34 mm mon$^{-1}$ before and after 1997, whereas wet years occur before 1997 with an
average of 51.03 mm mon$^{-1}$ but disappear after 1997.
3.2 Changes in summer precipitationshed and regional contributions
From 1980 to 2019, an average of 28.53 mm mon$^{-1}$ precipitation has been tracked
by the global surface, exceeding 93 % of IP summer precipitation with an average of
30.64 mm mon$^{-1}$. The climatology of the moisture contribution during the 40 years is
shown in Fig. 4 (a). The moisture contribution to IP generally decreases as its distance
to IP increases. Although the precipitationshed of IP summer precipitation is global in
scope, the contribution of the area far away is negligible to be considered. Therefore,
the 90$^{th}$ precipitationshed enclosed by the black line in Fig. 4 is given full attention as
the main moisture source region in the following text. The main moisture source of the
IP covers not only the local grids in the study region, but also several of non-local land
and oceanic areas. Due to the dominance of the westerlies in tropical–subtropical North





Atlantic corridor (Gimeno et al., 2010), as shown by the circulation in the Fig. 4(a),
most of the non-local source grids are located in the North American land and North
Atlantic Ocean to the west of the study area. The other source grids are located east of
North Atlantic Ocean and the IP, which is the downwind zone for water vapor transport,
covering Western Europe and the Mediterranean. Hence, the main moisture sources are
divided into the three partial regions of the local IP, the west and the east by the
boundary of the study area and the eastern boundary of the Atlantic Ocean (red and blue
lines in Fig. 4), and the contribution of each region to IP precipitation can be quantified
and compared.

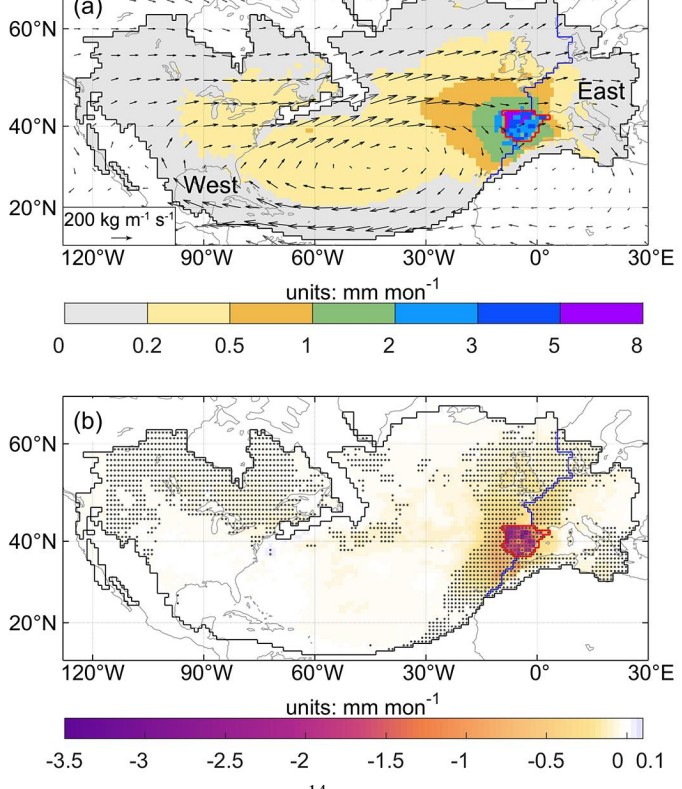






**Figure 4 (a)** Climatological 90th precipitationshed of the IP sink region and moisture contribution
to IP summer precipitation from 1980 to 2019. The black outlines show the 90th precipitationshed
boundary during the 40 years. The vectors represent the climatological monthly water vapor flux.
The red line encloses the study area, and the blue line divides the precipitationshed excluding the IP
into the west (left area) and the east (right area) regions. (b) Difference in moisture contribution in
the 90th precipitationshed between 1980-1997 and 1998-2019 (average of 1998-2019 minus average
of 1980-1997). The dots indicate 0.1 significance of the difference.

Affected by the transport distance, the grids with high contribution are located in

and around the target IP region, with the maximum values for grids in the northwest
corner of the IP. The local IP contributes 3.46 mm mon$^{-1}$ average summer precipitation,
with the precipitation recycling ratio of around 13.26 % during the 40 years. The west,
as the largest sub-region of the precipitationshed, contributes the most summer
precipitation of 19.38 mm mon$^{-1}$ and occupies 76.06 % of the tracked precipitation
averagely. While the east region, which is in an unfavorable downwind position in the
summer circulation, provides only 2.81 mm mon$^{-1}$ summer precipitation, accounting
for 10.68 %.

The difference in moisture contribution obtained from the 1998-2019 period minus

the 1980-1997 period is shown in Fig. 4(b). Almost all grid contributions show a
decrease after 1997. The grids with a large moisture contribution decline are mainly
concentrated in the IP, with the maximum reduction exceeding an average of 3 mm
mon$^{-1}$. Compared with other non-local source grids, the grids with higher contributions





along the east coast of the North Atlantic near the IP also have a slight but significant
reduction in contribution.

Due to the uneven distribution of grid contribution reduction in space, the area of

different percentile precipitationsheds differs in the two periods. The areas with
different colors in the distribution map of Fig. 5 represent the precipitationshed
boundaries at different percentiles in the two periods. During 1998-2019, the
precipitationshed boundary of each percentile extends westward in varying degrees
compared with those before 1997. The top decile of the contribution is still in the
western half of the IP. In the North Atlantic, the westward expansion of the western
boundary of the precipitationsheds is conspicuous, especially the 45th and 60th
percentile precipitationsheds shown in orange and green color in Fig. 5(a, b). This
westward extension implies that the significant and substantial reduction in the
contribution of the local grids and its surrounding grids results in a decrease in the
proportion of these areas. Therefore, for the same percentile of the precipitationshed,
only a smaller area concentrated by high-contribution grids is sufficient before 1997.
However, a larger area is required for the same proportion after 1997.



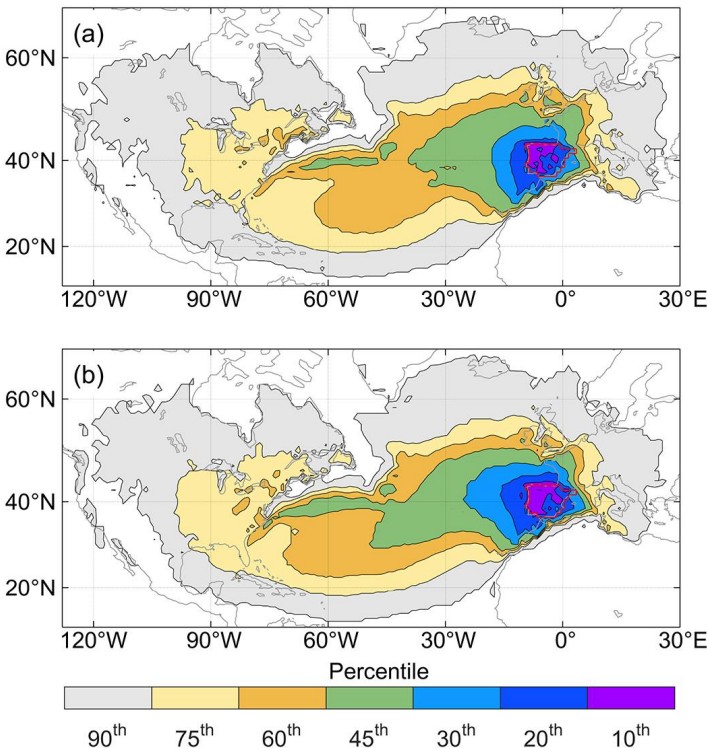


**Figure 5** Different percentile precipitationsheds during the two periods 1980-1997 (a) and 1998-

2019 (b).

Figure 6(a) shows the quantified precipitation contributed by the local IP, the west

and the east regions. The negative slopes in Fig. 6(a) indicate that the summer
precipitation contributed by these three regions has a downward trend, especially
significant for the IP and the west with slopes of -0.59 and -1.28 mm mon$^{-1}$ decade$^{-1}$.
These decreasing trends cause a 6.38 mm mon$^{-1}$ difference in precipitation from the
main source region in the two periods, which explain 82.64 % of the total reduction in
IP summer precipitation (7.72 mm mon$^{-1}$). In terms of the difference in the average



values of each region, the precipitation contributed by the local IP, the west and the east
significantly decreases from 4.38, 21.37 and 3.41 mm mon$^{-1}$ in 1980-1997 to 2.71,
17.76 and 2.32 mm mon$^{-1}$ in 1998-2019, respectively. 26.32 %, 56.53 % and 17.15 %
of the difference in main source supply between the two periods are due to the
contribution decline from the local IP, the west and the east, respectively.

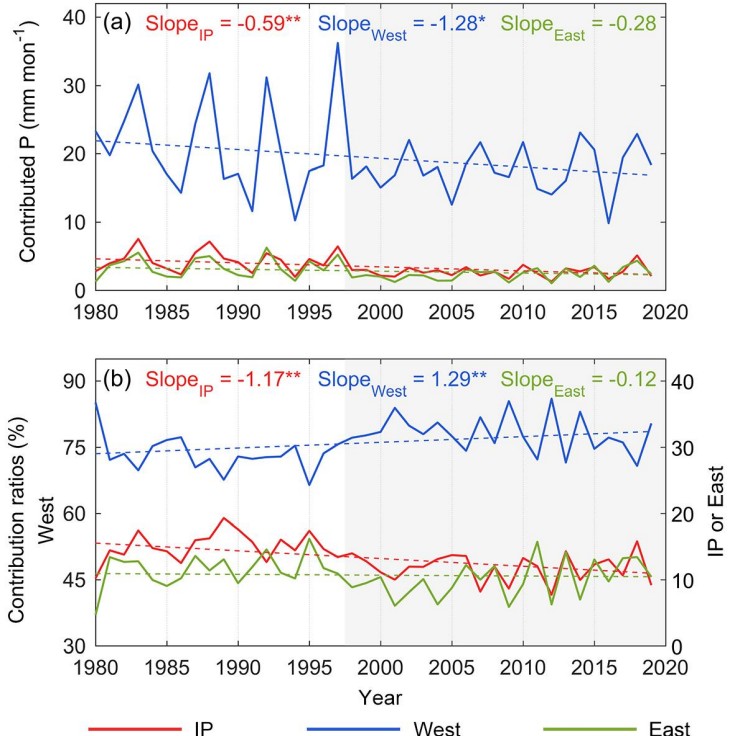


**Figure 6** Variations of contributed precipitation (a, unit of the slope: mm mon$^{-1}$ decade$^{-1}$) and
contribution ratios (c, unit of the slope: % decade$^{-1}$) from the IP, the west and the east region during
1980-2019 summer. '**' and '*' represent 0.05 and 0.1 level significance of the trend.

The variation and trend of the contribution ratio of each region are shown in Fig.

6(b). The proportion of contributions from the local IP and the east shows a decreasing



trend throughout the 40 years with the slope of -1.17 % decade$^{-1}$ and -0.12 % decade$^{-1}$,
which is consistent with the decreasing trends of their absolute contributions.
Conversely, although the precipitation contributed by the west shows a decreasing trend,
its proportion is significantly increasing and the slope is 1.29 % decade$^{-1}$. The average
contribution ratios of the local IP and the east decrease from 15.05 % and 11.49 %
before 1997 to 11.79 % and 10.02 % after 1997, while the ratio of the west increases
from 73.46 % to 78.19 %.
## 3.3 Differences in wet years and dry years
The dry years (1991, 1994, 2005, 2012 and 2016) and the wet years (1983 1987
1988 1992 and 1997) are selected as described in section 3.1. Of the two divided periods,
all the wet years only occur before 1997, while the dry years are distributed in both
periods with no decrease in its average value. This represents that although the average
summer precipitation after 1997 is reduced significantly compared with the previous
period, there is no decrease in the valley value of the precipitation series. Thus, the
disappearance of the wet years during 1998-2019 caused by the decrease of the
precipitation series peaks directly reflects the recent decrease in IP summer
precipitation.
During the entire 40 years, the difference in moisture contribution within the 90th
precipitationshed of IP summer precipitation between wet and dry years is shown in
Fig. 7(a). In the dry years, the significant reduction in the moisture contribution from





all grids in the main source region induces much lower precipitation than in the wet
years. On the grid scale, the larger declines primarily happened in the local IP, and the
grids with the largest drop, close to 9 mm mon$^{-1}$, are mainly concentrated in the west
and north of the IP. In each source region, an average of 6.41, 30.74 and 5.34 mm mon$^{-}$
$^1$ of summer IP precipitation is provided from the local IP, the west and the east in the
wet years, with 15.15 % recycling ratio, 72.19 % and 12.66 % contribution ratio. While
in the dry years, the average precipitation contributed from each region is 1.92, 11.66
and 1.40 mm mon$^{-1}$, accounting for 12.93 %, 77.70 % and 9.37 %, respectively. All
three regions contribute more to summer precipitation in wet years than in dry years,
and compared with dry years, the contribution ratios of the local IP and the east in wet
years are also higher. The disappearance of wet years during 1998-2019 further
motivates similar changes between the two periods. The decrease in the frequency of
wet years with higher local recycling ratio and higher contribution ratio of the east leads
to an increase in the proportion of the summer precipitation originating from the
remaining other region, namely the west, during the same period.

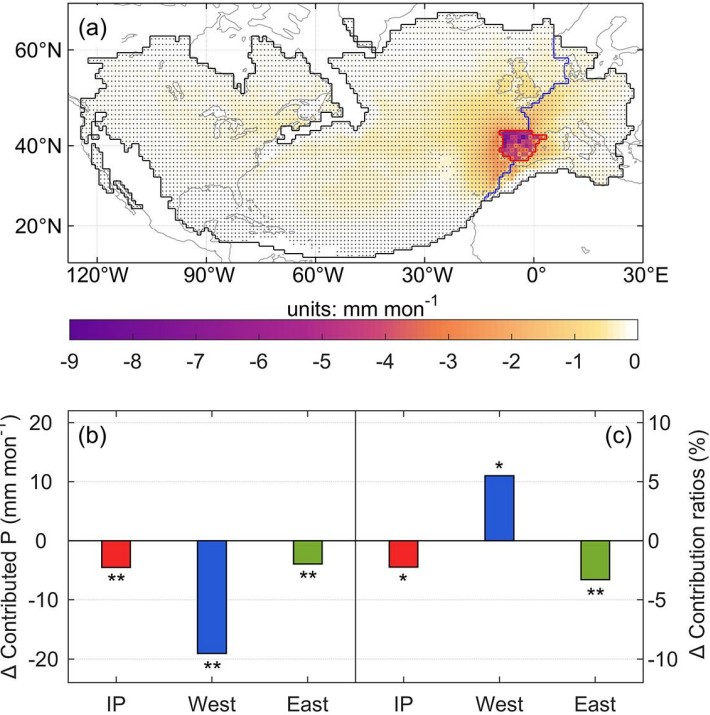


**Figure 7** (a) Difference in moisture contribution in the 90[th] precipitationshed between the dry years

and the wet years (average of dry years minus average of wet years). The dots indicate 0.1

significance of the difference. The changes in average precipitation contributed from each region

(b) and their average contribution ratios (c) between the dry years and the wet years. '**' and '*'

represent 0.05 and 0.1 level significance of the difference.

The dry years in the two periods have been divided and compared with each other,

and the differences between the two periods are shown in Fig. 8. From the distribution

of differences, the grids with reduced moisture contribution are mainly located in the

IP and the east region, and the southern part of the IP has the largest decrease (Fig. 8(a)).

Mainly dominated by these negatively changing grids, both the absolute contribution




and the contribution ratio of the local IP and the east have dropped significantly, with
0.53 and 0.42 mm mon$^{-1}$ decrease in contributed precipitation and 3.58 % and 2.81 %
contribution ratio reduction, respectively (Fig. 8(b, c)). For the west region, however,
it raises the moisture contribution to the summer precipitation by 1.22 mm mon$^{-1}$ in dry
years after 1997, causing a 6.39 % increase in its contribution ratio. Despite the dry
years with no decrease precipitation between two periods, the decrease in local
recycling is still noticeable.

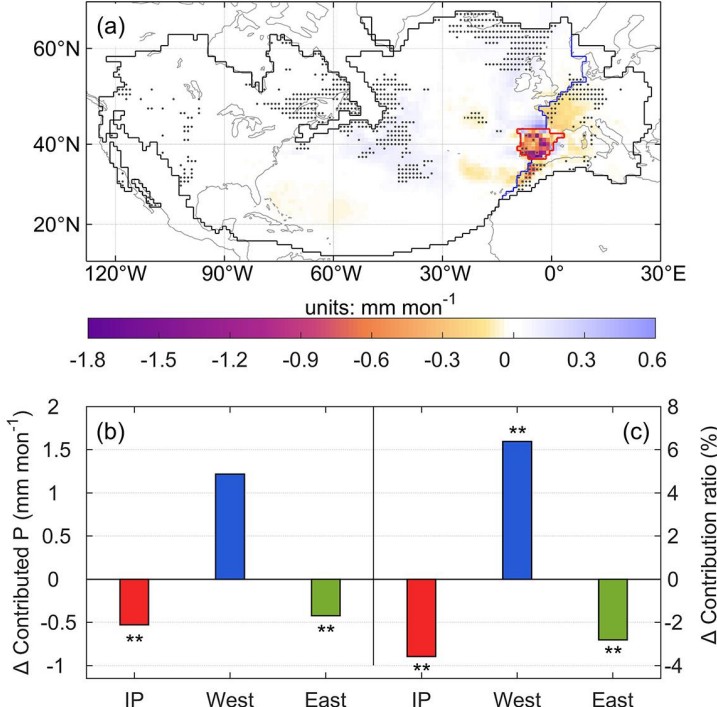


**Figure 8** (a) Difference in moisture contribution in the 90[th] precipitationshed in the dry years
between 1998-2019 and 1980-1997. The dots indicate 0.1 significance of the difference. The
changes in average precipitation contributed from each region (b) and their average contribution





ratios (c) in the dry years between 1998-2019 and 1980-1997. '**' and '*' represent 0.05 and 0.1
level significance of the difference.

## 4. Discussion

The trends in the contribution from the three source regions, the local, the west
and the east regions, to all seasonal and annual precipitation over the past 40 years are
listed in Table 1. In general, the decreasing trend maintained by the local IP and the east
region are closely related to the drought in the Mediterranean basin (Ribeiro et al., 2020;
Russo et al., 2019), and the increasing proportion of the west can be explained by the
increasingly important role of the oceanic moisture in terrestrial precipitation (Gimeno
et al., 2020; Vicente-Serrano et al., 2018). The simultaneous decrease in the moisture
contribution from all three regions is responsible for the significant decrease in only the
summer precipitation series among all seasonal or annual precipitation. In particular,
the local recycling ratio in summer is obviously way down, differentiating the reduced
summer precipitation from the other seasons. It is worth highlighting that this
significant decrease in recent summer precipitation over the IP in this study is based on
a short record (1980-2019) from ERA5, while a long-term assessment of precipitation
(1850-2018) from multiple sources still lacks a statistically significant decreasing trend
(Peña-Angulo et al., 2020). Nevertheless, the changes in the recent four decades still
show the significant influence of the local recycling, especially on the trend of summer
precipitation and variation of summer wet and dry years.



**Table 1** Trends of contributions from the IP, the west and the east to annual and seasonal
precipitation, and the trends of their contribution ratios.

| | Contributed precipitation (mm mon$^{-1}$ decade$^{-1}$) | | | | | Contribution ratio (% decade$^{-1}$) | | | | |
| --- | --- | --- | --- | --- | --- | --- | --- | --- | --- | --- |
| | Annual | Spring | Summer | Autumn | Winter | Annual | Spring | Summer | Autumn | Winter |
| IP | -0.24** | -0.30 | -0.59** | -0.03 | -0.03 | -0.49** | -0.66** | -1.17** | -0.14 | -0.08 |
| West | 0.53 | 1.67 | -1.28* | 1.23 | 0.52 | 0.81** | 0.80 | 1.29** | 0.38 | 0.77 |
| East | -0.17 | -0.06 | -0.28 | -0.05 | -0.29 | -0.32 | -0.14 | -0.12 | -0.24 | -0.69 |

'**' and '*' represent 0.05 and 0.1 level significance of the trend.

The remarkable decrement of summer precipitation can be attributed to the

simultaneous and large reduction of contributions from all three source regions. The
strong land-sea contrast caused by the warming land surface makes the advected air
mass from Atlantic experience drying (Cramer et al., 2018; Kröner et al., 2017),
resulting in a decrease in the moisture contribution from the Atlantic Ocean in the west
to the IP precipitation. In addition, the extension of Hadley circulation makes the IP
more strongly affected by subsidence with higher static stability and lower frequency
of extreme heavy precipitation (Brogli et al., 2019). However, the ocean warming
patterns and thermodynamics can promote precipitation in cold seasons (Brogli et al.,
2019), just as shown by the increasing contributed precipitation from the west in
autumn and winter in Table 1. It suggests the drivers leading to less summer
precipitation do not generally cause a similar change in precipitation in the other
seasons.

As an important indicator to describe the interaction between the surface and



atmospheric processes, the change in precipitation recycling ratio takes into account
changes in both precipitation and the contribution of local evaporation (Goessling and
Reick, 2011). For the IP, its significant reduction in local moisture contribution is most
likely due to the weakening of local evaporation (Fig. 9). Due to the positive correlation
between soil moisture and precipitation in summer, the declining precipitation leads to
the shortage of soil water supply, the limitation of soil water evaporation capacity and
the consequent reduction in surface evaporation (García-Valdecasas Ojeda et al., 2020;
Ruosteenoja et al., 2018). Especially in summer, when the soil moisture and recycling
process driven by evaporation are regarded as an active source of moisture (Jung et al.,
2010; Vicente-Serrano et al., 2014) , this weakening of the local moisture recycling
again leads to a decrease in precipitation. This continuous feedback of the interactions
of soil moisture evaporation and precipitation can exacerbate the water resource
depletion and summer drought.

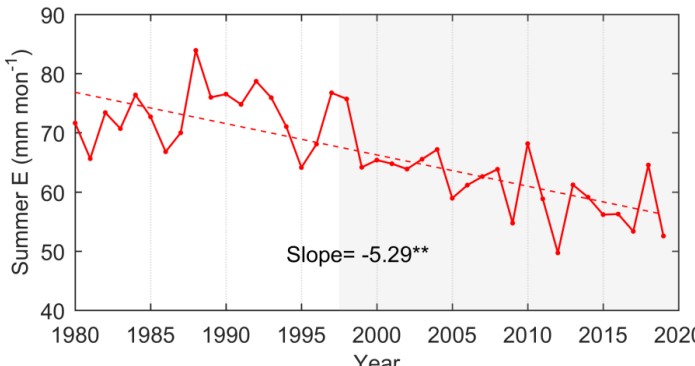


**Figure 9** Time series of IP summer evaporation from ERA5 during 1980-2019 (unit of the slope:
mm mon$^{-1}$ decade$^{-1}$). '**' represents 0.05 level significance of the trend.




## 5. Conclusions

In this study, using the reanalysis data ERA5 and WAM-2layers model, we investigated how changes in moisture contribution from the source affect the reduction in summer precipitation between 1980-1997 and 1998-2019. The major findings are summarized below.

1) The reduction of contribution to IP summer precipitation is mainly concentrated in the IP and its neighboring grids. The local IP grids show the greatest reduction, and the surrounding grids show a slight but significant decrease.

2) Compared with the period of 1980-1997, the decrease in the moisture contribution from the IP, the west and the east during 1998-2019 results in the reductions of 1.7, 3.6, and 1.1 mm mon$^{-1}$ of the IP precipitation, accounting for 26 %, 57 %, and 17 % of the main source supply reduction, respectively.

3) The contributions from the local IP and the east keep declining during the 40 years. In particular, the significant reduction in local recycling, reflected in the disappearance of wet years after 1997 and the reduction of local contributions in dry years, suggests a close link with the decrease in summer precipitation.

**Code and Data availability**

Code and data used in this manuscript are available from the corresponding author upon a reasonable request.



**Author contributions**

MG and QT designed the study; YL performed the analysis and calculation; CZ contributed to the application of the model in this study; YL prepared the manuscript draft, and all co-authors reviewed and edited the manuscript.

**Competing interests**

The authors declare no competing interests.

**Acknowledgements**

This study was partly funded by the National Natural Science Foundation of China (41730645) and the Strategic Priority Research Program of Chinese Academy of Sciences (XDA20060402). The authors would like to thank the EU and Innovation Fund Denmark (IFD) for funding within the framework of the FORWARD collaborative international consortium financed through the ERA-NET co-fund WaterWorks2015 integral part of the 2016 joint activities developed by the "Water Challenges for a Changing World" joint programme initiative (Water JPI).



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
