# Peer review of "Recent decrease in summer precipitation over the Iberian Peninsula"

_Hydrology and Earth System Sciences, 2021_

## Author Comment (AC1)

**Response to the comment of Referee #1**

This manuscript of Liu et al. presents and discusses decreases in summer precipitation over the Iberian Peninsula (IP) by analyzing the moisture sources. They applied WAM-2layers to track moisture sources that contribute to the summer precipitation. The moisture source region is divided into three sub-regions: western, IP, and eastern regions. Their findings show how moisture contribution changed from the three regions resulted in less precipitation over the IP.

The results showed by Liu et al. help to better understand what the changes and contributions of the three moisture sources are and how they affect the IP summer precipitation. The motivation is generally interesting, and the figures and text are clear. The method and analysis are clearly stated and explained. I have a few questions and recommend the manuscript for publication after a minor revision

We gratefully thank the reviewer for the time and effort spent making these constructive remarks and thoughtful comments. These comments can significantly enable us to improve the manuscript. Below, each of the reviewer's comments have been replied point by point.

**Questions/comments:**

1. About the identified west region, the moisture from this region was considered as a contribution from the circulation over the North Atlantic Ocean. North America is also included in this region. However, in Figure 4b it seems that North America is

not connected to the other moisture source regions (e.g., the eastern North Atlantic Ocean) when you consider the significant differences between wet and dry years. What are the physical mechanisms of moisture transport from North America? Is it reasonable to include North America in the west region?

Response: We are appreciative of this question raised by the reviewer. In this study, we include North America in the west region for the following reasons. Firstly, we do not think North America can be considered as a separate source region. It is true that due to the long distance of North America from the Iberian Peninsula, the climatological contribution of North America land evaporation to precipitation in the study region is much smaller compared to other regions (most areas contribute less than 0.2mm mon$^{-1}$, and a small part contribute 0.2-0.5 mm mon$^{-1}$, as shown in Figure4(a)) but still contributes (technically speaking, everywhere contributes although the region far away from Iberian Peninsula contributes much less). Therefore, it has not been considered as a main source of water vapor in the Iberian Peninsula in many studies or its contribution has not been considered enough to be discussed independently (Gimeno et al., 2010; Winschall et al., 2014).

Secondly, as we aimed to delineate a large enough precipitationshed to trace IP precipitation as completely as possible, a part of North America will, inevitably, be taken into consideration in our study. This part is influenced by the North Atlantic anticyclonic structure in terms of the mechanisms. Besides, it and the North Atlantic at similar latitude together form a relatively stable atmospheric basin in summer according

to the construction of global atmospheric moisture networks (Zhang et al., 2020). For these reasons, we believe that it is reasonable to include the contribution of North America in the west region in this study without ignoring it. And we will add these descriptions of the basis for sub-source division in the revision.

*References:*

*Gimeno, L., Nieto, R., Trigo, R. M., Vicente-Serrano, S. M., and López-Moreno, J. I.: Where does the Iberian Peninsula moisture come from? An answer based on a Lagrangian approach, J. Hydrometeorol., 11, 421-436, https://doi.org/10.1175/2009JHM1182.1, 2010.*

*Winschall, A., Sodemann, H., Pfahl, S., and Wernli, H.: How important is intensified evaporation for Mediterranean precipitation extremes?, Journal of Geophysical Research: Atmospheres, 119, 5240-5256, https://doi.org/https://doi.org/10.1002/2013JD021175, 2014.*

*Zhang, Y., Huang, W., Zhang, M., Tian, Y., Wang, G., and Zhong, D.: Atmospheric Basins: Identification of Quasi-Independent Spatial Patterns in the Global Atmospheric Hydrological Cycle Via a Complex Network Approach, Journal of Geophysical Research: Atmospheres, 125, e2020JD032796, https://doi.org/https://doi.org/10.1029/2020JD032796, 2020.*

2. As for the east region, it seems that it is at the downwind location of the circulation over the North Atlantic Ocean. It is not clear in the manuscript how the moisture from the east region contributes to the IP precipitation. It would be helpful if the authors could clarify the physical mechanism.

Response: Thank you for your comment. Normally, the east region sits downwind of the circulation. However, as the wind can blow everywhere due to pressure variation, the moisture transport trajectories are more complexed than the climate mean 2-

dimentional figure (Fig.4a) tells. Low pressures can form occasionally in the IP, which bring the moisture from the downwind east to the IP.

We will add these relevant explanations to the revision: The east region includes the Mediterranean Sea where the atmospheric moisture divergence is positive almost everywhere, indicating a net water flux from it to the atmosphere (Mariotti et al., 2002). The evaporation from the Mediterranean Sea moistens the air parcels and flows towards the surrounding land, and becomes the main short-term moisture source regions affecting the IP, especially the eastern IP (Gimeno et al., 2010; Vázquez et al., 2020). On the eastern peninsula, the most important pattern for producing precipitation extreme events is characterized by the presence of a cutoff low at mid-levels, mostly in autumn, together with an easterly moisture flow from the Mediterranean Sea, generating significant instability over the area (Merino et al., 2016). Although the eastern region is located downstream of the dominant wind direction, some of the large amount of evaporation it provides will contribute to precipitation over the adjacent IP land in a short time by the process of water vapor exchange, especially for the frequent convection precipitation in summer.

*References:*

*Gimeno, L., Nieto, R., Trigo, R. M., Vicente-Serrano, S. M., and López-Moreno, J. I.: Where does the Iberian Peninsula moisture come from? An answer based on a Lagrangian approach, J. Hydrometeorol., 11, 421-436, https://doi.org/10.1175/2009JHM1182.1, 2010.*

*Mariotti, A., Struglia, M. V., Zeng, N., and Lau, K.-M.: The Hydrological Cycle in the Mediterranean Region and Implications for the Water Budget of the Mediterranean Sea, Journal of Climate, 15, 1674-1690, https://doi.org/10.1175/1520-0442(2002)015<1674:Thcitm>2.0.Co;2, 2002.*

Merino A, Fernández-Vaquero M, López L, Fernández-González S, Hermida L, Sánchez JL, García-Ortega E, and Gascón E (2016) Large-scale patterns of daily precipitation extremes on the Iberian Peninsula. *International Journal of Climatology* 36

Vázquez, M., Nieto, R., Liberato, M. L. R., and Gimeno, L.: Atmospheric moisture sources associated with extreme precipitation during the peak precipitation month, *Weather and Climate Extremes*, 30, 100289, https://doi.org/https://doi.org/10.1016/j.wace.2020.100289, 2020.

3. Evaporation is used to explain the local moisture contribution. Although both have a decreasing trend, how does the annual evaporation (in Figure 9) correlate to the contributed P in Figure 6 in terms of interannual variability?

Response: Thanks for pointing this out. We agree that it is necessary to show correlations between local recycling and local evaporation. We will calculate the Pearson correlation coefficients between evaporation and locally contributed precipitation and its recycling ratio respectively, and add the result to show the correlations between these interannual variabilities.

4. Comments: The role of local recycling is stressed in the conclusion. However, looking at Figure 6a and Table 1, it seems to me that the changes in contribution from the west region have the most dominant effect. I think this should be made clear in the conclusions.

Response: Thanks for your rigorous consideration. Among the three subregions, the most dominant effect of the west region, as you mentioned, is more due to its wide coverage with a large number of grids, which makes the cumulative amount of the entire region the largest. We will rephrase conclusion appropriately to clearly emphasize the influence of the west region.

**Technical corrections**

1. The grey shading in Figures 2,6 is not defined in the captions.

Response: Thanks for your comment. We will add the explanations to the captions of Figure 2 and 6 in the revision. Due to the abrupt change in the summer precipitation sequence in 1997, the 40 years were divided into two periods in our study. In order to visually show the fluctuation and change of the whole 40 years and the two separated periods, we used different background colors in the figures. The white is for the period of 1980-1997, and the grey shading is for 1998-2019 period.

---

## Author Comment (AC2)

**Response to the comment of Referee #2**

The article 'Recent decrease in summer precipitation over the Iberian Peninsula (IP) closely links to the reduction of local moisture recycling' by Liu et al. describes the decrease in summer precipitation over the IP and how moisture source regions contribute to this change. First, the temporal variation in the annual average of precipitation is studied for the years 1980-2019. Observations are used to validate ERA5 precipitation for this region. Second, a mutation analysis is used to find a significant mutation in precipitation over the years 1980-2019. Next the study focusses on two major time periods. Period (1) 1980-1997, and period (2) 1998-2019. This distinction is based on the results of the mutation analysis. Thereafter, changes in precipitation and precipitationsheds from the first time period to the second period are studied. Precipitationsheds are calculated using WAM2-Layers. The precipitationshed is split into three regions. The local region, including the IP, the western region and the eastern region. Next, the temporal variation in the contribution of these regions to precipitation in the IP is studied. Last, a distinction is made between wet and dry years in the entire time period. In this step, firstly, the contribution of the three different sources to precipitation in wet and dry years throughout the entire period is studied. Secondly, the change in contribution to precipitation in the IP of the three sources in dry years between 1980-1997 and 1998-2019 is studied.

The major conclusions are (1) the reduction in IP is the largest, neighboring grid cells show a smaller decrease, (2) in summer, there is a reduction in moisture contributed

from the IP (26%), the west region (57%), and the east region (17%), and (3) the reduction in local recycling closely links to a decrease in summer precipitation. These results can help us to better understand changes in precipitation over the study region.

I believe this article fits the scope of the Journal 'Hydrology and earth system sciences'. The results help us to understand the hydrological system of the IP better as the authors analyzed spatial and temporal characteristics of water resources of their study region. By splitting the precipitationshed up in different source areas we can get a better understanding of the role of different physical processes in the cycling of continental water.

I believe this is an interesting and relevant article, that fits to the scope of this journal. However, revisions are necessary before I recommend this article for publication.

We thank the reviewer's time and the constructive feedback to help us improve this manuscript. These comments, suggestions and the corresponding responses are listed below, and indicate how we plan to revise our manuscript.

**Strengths**

I believe the goal, methods, and results of this study are clearly described and presented. I find this a very interesting study. The figures are very informative and clear. I only have some minor comments about the figures and captions (See minor comments). Furthermore, I like the table in the discussion as it gives a nice clear overview of the change in contribution of the three source regions.

**Main points of improvement**

First, this study uses data with a spatial scale of 1x1 degree. I was wondering why the authors decided to use data with a resolution of 1 degree in this study. WAM-2layers is mostly run at a spatial scale of 1.5 degree (e.g. Link et al., 2020). Tuinenburg and Staal (2020) showed that the time step and grid cell size can influence the output of the normally could happen at high latitudes, the Courant number could become larger than 1. If the Courant number is larger than 1, moisture cannot be correctly transported over a Eulerian grid (Tuinenburg and Staal, 2020). The authors decided to use a smaller spatial resolution than 1.5 degree, which will affect the Courant number. Therefore, I would be interested in a clarification of the authors on if the change in spatial resolution affects the output of their model.

Response: Thanks for your thoughtful comments. The time step and grid cell size can definitely affect the stability of WAM-2layers. For the widely used 1.5-degree latitude-longitude spatial scale, the input data is usually interpolated 15 (Keys et al., 2014) or 30 mins (van der Ent and Savenije, 2011; van der Ent et al., 2010) to keep the Courant number less than 1, and the grids at high latitude (south of 57°S and the grids north of 80°N) are do not participate in the calculation due to their severe instability (van der Ent et al., 2010). With the appropriate time step, the calculation speed and storage space occupied when running WAM-2layers with a 1.5-degree data is widely acceptable. However, this does not mean that this model cannot be used with other spatial resolution data. When spatial resolution of the data is changed, the time step should be considered

to ensure that the Courant number is less than 1, and the time and storage space will also be affected and increase exponentially. E.g., Benedict et al. (2021) used 0.25° ERA5 data with 6 mins time step. In addition, it is common to use 1° data for WAM-2layers calculation. The 1° data of ERA-Interim (Zhang C. et al., 2019), TRMM (Zhang C., 2020) and JRA-55 (Li et al., 2019) are used after the time step is reduced to 0.25h to make sure that the Courant number is less than 1 and the numerical stability of WAM-2layers. In our study, we choose to use the new updated ERA5 data also with 1 degree which can cover our study region (the IP) very well and the time step is also set as 0.25h to keep the stability of WAM-2layers.

*References:*

Benedict, I., van Heerwaarden, C. C., van der Linden, E. C., Weerts, A. H., and Hazeleger, W.: Anomalous moisture sources of the Rhine basin during the extremely dry summers of 2003 and 2018, Weather and Climate Extremes, 31, 100302, https://doi.org/https://doi.org/10.1016/j.wace.2020.100302, 2021.

Keys, P. W., Barnes, E. A., van der Ent, R. J., and Gordon, L. J.: Variability of moisture recycling using a precipitationshed framework, Hydrology and Earth System Sciences, 18, 3937-3950, https://doi.org/10.5194/hess-18-3937-2014, 2014.

Li, Y., Su, F., Chen, D., and Tang, Q.: Atmospheric Water Transport to the Endorheic Tibetan Plateau and Its Effect on the Hydrological Status in the Region, Journal of Geophysical Research: Atmospheres, 124, 12864-12881, https://doi.org/https://doi.org/10.1029/2019JD031297, 2019.

van der Ent, R. J., and Savenije, H. H. G.: Length and time scales of atmospheric moisture recycling, Atmos. Chem. Phys., 11, 1853-1863, https://doi.org/10.5194/acp-11-1853-2011, 2011.

van der Ent, R. J., Savenije, H. H. G., Schaefli, B., and Steele-Dunne, S. C.: Origin and fate of atmospheric moisture over continents, Water Resour. Res., 46, https://doi.org/10.1029/2010WR009127, 2010.

Zhang, C.: Moisture source assessment and the varying characteristics for the Tibetan Plateau precipitation using TRMM, Environmental Research Letters, 15, 104003, https://doi.org/10.1088/1748-9326/abac78, 2020.

Zhang, C., Tang, Q., Chen, D., van der Ent, R. J., Liu, X., Li, W., and Haile, G. G.: Moisture Source Changes Contributed to Different Precipitation Changes over the Northern and Southern Tibetan Plateau, Journal of Hydrometeorology, 20, 217-229, https://doi.org/10.1175/jhm-d-18-0094.1, 2019.

Second, I believe that the physical processes that drive the transport of moisture to the IP should be highlighted a bit better. It would be very interesting if it is possible to support the decrease in the contribution of the different regions with a change in these physical processes, or the importance of these physical processes. This is done for the contribution of the western region. The authors describe how an increase in surface temperature over land reduces rainfall. I am interested in such an explanation for the other two source regions as well. In the discussion, the authors use the change in summer evaporation to elaborate a bit on this point as well. However, while reading this part, I was wondering how the summer evaporation relates specifically to the dry years. Perhaps the authors could elaborate on this. In addition, Table 1 shows the important contribution of a change in the western source for summer as well. Perhaps the authors could clarify, using the physical mechanisms, why the focus in this article is on the local moisture recycling and not on the western source region.

Response: Thanks for your rigorous consideration. Firstly, we will add a part of discussion to explain the physical processes related to moisture transport forcing the IP precipitation reduction for the west and east region. Secondly, for the present discussion about the influence of local evaporation on the decreasing recycling, we will add the more content to specify the relations and focus more on the dry years. Thirdly, Among the three subregions, the most dominant effect of the west region, as you mentioned, is more due to its wide coverage with a large number of grids, which makes the cumulative

amount of the entire region the largest. We will rephrase conclusion appropriately to clearly emphasize the influence of the west region.

Third, I think the authors should highlight the relevance of the research a bit better. I believe this manuscript would be more informative if the authors clearly describe the implication of their results. The scientific relevance is described as follows: a better understanding of the summer precipitation decline in the IP. However, I miss what this additional understanding could be used for. Please, describe this relevance in the discussion and shortly mention it in the conclusion.

Response: Thanks for your suggestion. We decide to re-emphasize the scientific relevance of this study as follows: Decreasing summer precipitation over the IP could lead to escalation of drought, especially with its high temperature and low rainfall characteristics of Mediterranean climate. This precipitation anomaly in the sink is directly linked to the source changes, so we attempted to attribute the decreasing precipitation to the changes in the evaporation contribution from the main source regions. Although the importance of the ocean as a moisture source is always emphasized, our results underscore the important consistency of changes in the contribution and proportion of local moisture contribution and the reduction of IP precipitation. It can provide a scientific reference for the prediction and management of droughts that may be caused by reduction of precipitation from the perspective of

moisture contribution and sources. We will restate this in the introduction and also mention it in the discussion and conclusion.

Finally, I was wondering why the authors call the recycling of moisture within the IP local recycling and not regional recycling as the moisture recycles within a region. The reader could misunderstand the spatial scale of this study when reading local moisture recycling as local scale could be understood as a smaller spatial scale than the spatial scale of the IP. This misunderstanding can be prevented when the authors state their definition of local recycling in the introduction, or when they use the terminology regional recycling. The latter would prevent misunderstanding the title.

Response: We agree that moisture recycles within a region. The 'local' moisture recycling means the regional moisture cycle close to the target area, distinguishing from moisture from a non-local or remote region. The term 'local moisture recycling' has been widely used in many studies. To avoid misunderstanding, we will clarify the 'local moisture recycling' in the introduction. We do not use 'local recycling', instead we use 'local moisture recycling' in the revision.

**Minor points of improvement**

Minor points concerning figures and tables

1. In multiple captions the letter indicating which plot a specific part in the caption is about, is sometimes located in front of the subject and sometimes behind the subject. Please, change this so all captions are consistent. I.e. either the letter in front or behind the part of the caption it refers to.

Response: Thanks for your suggestion. We will make it consistent: The letter will be placed in the sentence and before the part of the caption it refers to.

2. For figure 2, the y-axis for some of the plots starts at zero and for some of the plots it does not. However, I believe it would be easier to compare the result if the y-axis of all plots start with zero. Could the authors please consider this?

Response: We will replot the figure with the same y-axis starting from 0 for better comparison.

3. For figure 3, 4, 7 and 8, the colorbar indicates the unit of the quantity the authors plot here. Please also indicate what quantity is plotted above the colorbar. This will make the plots easier to read.

Response: We will add the name of quantity to the figure.

4. For figure 5, could the authors please make a small change in the caption? Please clarify it is the precipitationshed of the IP.

Response: Thanks for your kind remind. We will change that.

5. For figure 6, it took me some time to understand plot (b) has two y-axes. This could be clarified by changing the color of the left y-axis to blue.

Response: Thanks for your suggestions. We will change the color of left y-axis and y label to blue in figure 6.

6. In the caption of Figure 7, the authors indicate how they calculated the difference between wet and dry years. However, for Figure 8 the authors did not include such a description. Please add this to the caption of Figure 8.

Response: We will add that.

General minor points

1. In the abstract (line 19) the authors mention the source. By using 'the source' it seems like there is only one source. Please rephrase to sources or mention the three main sources that are studied in this research.

Response: Thanks for your suggestion. We will change "the source" to "the sources".

2. In the introduction (line 33), the authors state the IP is located in the Mediterranean area. However, there are five major Mediterranean areas around the world. Please rephrase to the Mediterranean basin which is one of these five Mediterranean areas.

Response: Thanks for your advice and we will change that in revision.

3. In the description of the study areas the authors describe the topography with the word high (line 93). Do the authors refer to elevation here? If so, please clarify by using a word like elevated.

Response: Yes, we want to express the high altitude. We will change the word to elevated in the revision.

4. In Equation (1) the authors use a +/- sign in front of the vertical moisture transport term. I can imagine this is because of the direction of the transport. However, if it is negative isn't the minus already implemented in the F term?   In addition, when reading this article I was wondering what the residual term presents. Please clarify both points.

Response: $F_V$ term is Equation (1) is the vertical moisture transport between the bottom and top layer. Generally, we think that the net vertical flux for one grid is only positive or negative. However, Ruud, the developer of WAM-2layers, pointed that only considering net $F_V$ on one direction is too small, which can be attributed to the turbulent moisture exchange. So during WAM-2layers calculation, it has to use a vertical flux of 4Fv in the direction of the net flux and 3Fv in the opposite direction. Therefore, both + and - is maintained in the equation (van der Ent et al., 2014) . The difference between the 'real' and the 'model' moisture storage is called the residual term α, which is used to make sure the water balance. It is resulted from data assimilation in the ERA5 data and that the offline tracking scheme calculates the water balance on a coarser spatial and temporal resolution (van der Ent et al., 2014).

*Reference*

*van der Ent, R. J., Wang-Erlandsson, L., Keys, P., and Savenije, H.: Contrasting roles of interception and transpiration in the hydrological cycle - Part 2: Moisture recycling, Earth Syst. Dynam., 5, https://doi.org/10.5194/esdd-5-281-2014, 2014.*

5.  In Equations (2) and (3) the authors use the sign for a cross product. However, do the authors want to indicate a cross product or a product?

Response: Sorry for the misleading and incorrect expression. We want to indicate a product and we will correct these equations in the revision.

6. The sentence of lines 150-152 is a bit unclear to me. Could the authors rephrase this sentence?

Response: We will rephrase it.

7. The mutation test and the term mutation are new to me. Reading the article I understand this test can be used to find a significant change within data. Please include an explanation on this in one sentence (or a few) around line 159.

Response: We add simple descriptions to make it easier for readers to understand.

8. Line 191, the authors mention both Iberia01 and ERA5 show statistically significant changes in 1997. I would like to ask the authors to indicate this with a number resulting from their mutation analysis.

Response: Thanks for your suggestion and we will show the result of the specific value.

9. In line 211, the authors use the words 'are separated'. This use of words is a bit unclear to me. Do the authors mean that the dry years stand out?

Response: What we want to express is that according to the above definitions of dry or wet years, the selected wet years only exist in the first period (before 1997), while dry

years are distributed in both two periods. Therefore, all selected dry years are also divided into two time periods for the following comparison.

10. In lines 215-217, the sentence is a bit unclear to me. I believe the authors state that the average precipitation between 1980 and 2019 is 28.53 mm per month and that for these years the summer precipitation in the IP is on average 30.64 mm per month. This indicates that the summer precipitation is above average. However, Mediterranean areas are wet in winter and dry in summer. Therefore, this result seems to be unexpected. Please elaborate on this.

Response: Sorry for the misleading. Due to the atmospheric retention of moisture and the residual item of the model, not all summer precipitation (30.64 mm mon$^{-1}$ averagely) can be traced back to the evaporation source within the backtracking period (1 month in this study). Here, 28.53 mm mon$^{-1}$ is not the average annual precipitation. It indicates there is summer precipitation (28.53 mm per month) has been traced to the sources. We will rephrase this sentence to make it more clear in the revision.

11. In line 254, could the authors indicate the percentage of the reduction of 3 mm per month?

Response: Thanks for your suggestion and we will add the percentage value.

12. In lines 323-324, the authors state that the disappearance of wet years in the second time period motivates changes similar to a higher contribution of all three areas to precipitation in the IP. This statement is a bit vague to me. Perhaps because the authors use the word motivate. Do the authors mean to say that they see a similar pattern? If so, could the authors please rewrite this sentence?

Response: Yes, and we will rewrite it.

13. In the first lines of the conclusion, the authors state 'moisture contribution from the source'. However, in their study the authors focused on three source regions. I believe the conclusion would improve if the authors mention the three sources here. This nicely links to point 2 in the conclusion where the authors mention the regions. If these regions are already introduces in the beginning of the conclusion, this part might become easier to read.

Response: Thanks for your suggestion. We will first mention the three source regions to summarize more clearly later in the manuscript.

Minor points in use of grammar

1. The sentence in lines 71-74 is a bit unclear to me due to the sub sentence in the middle. I had to read it multiple times to understand it properly. Please rephrase.

Response: We will rephrase it.

2. Throughout the manuscript the authors miss the word 'the' in several sentences. Some examples: line 95: the northwest to the southeast, line 104: the ERA5 dataset. Could the authors please take a close look at their full text for this?

Response: We will conduct a careful full-text check on this.

3. In line 116 the authors use the word avoid. However, it is difficult to avoid uncertainty. I believe the authors do this check to account for uncertainty, or get an idea of the uncertainty in their study. I believe it would be better to use a different word than avoid. Could the authors please consider this?

Response: Thanks for your suggestion and we will rephrase it.

---

## Author Response (AR2)

**Comment:** The referee comments were well addressed and the manuscript has improved in consequence. However, I agree with referee #2 that the sliding t-test and accompanying equations (4-6) are not sufficiently explained. The equations appear rather clunky, x should be written x_t (where _ denotes an underscore) and a reference to literature is missing. Is your method similar to [De Souza, 1983]? Then you might consider adopting his notation with x_i and y_i.

*Reference:*

*Peter de Souza (1983): Edge detection using sliding statistical tests, Computer Vision, Graphics, and Image Processing, 23(1),1-14, https://doi.org/10.1016/0734-189X(83)90051-8.*

**Reply:** Thanks to the editor for pointing this out. In the revision, we have simplified the complicated equations (4-6) into one equation (line 175). And variables are distinguished by subscripts as you suggested. The sliding t-test used in this study is very similar to the reference you mentioned, but slightly different. And one more closely related reference has been cited in the revision (Maasch, 1988).

*Reference:*

*Maasch, K. A.: Statistical detection of the mid-Pleistocene transition, Climate Dynamics, 2, 133-143, https://doi.org/10.1007/BF01053471, 1988.*